# Integrating Complementary and Alternative Medicine into General Practice Training: A Regional Survey in South Tyrol

**DOI:** 10.3390/healthcare13070797

**Published:** 2025-04-02

**Authors:** Christian J. Wiedermann, Giuliano Piccoliori, Adolf Engl

**Affiliations:** Institute of General Practice and Public Health, Claudiana—College of Health Professions, 39100 Bolzano, Italy

**Keywords:** complementary and alternative medicine, general practitioners, medical education, South Tyrol, CAM integration, primary care

## Abstract

**Background/Objectives:** Complementary and alternative medicine (CAM) is increasingly being recognized as an important component of primary care; however, its integration into medical education and practice remains inconsistent. This study explores the attitudes, practices, and educational needs of young career general practitioners in South Tyrol, a linguistically and culturally diverse region. **Methods:** A questionnaire was distributed to all 131 general practitioners currently in training or who completed their specialization within the last 10 years, and 37 responses were analyzed. **Results:** The findings revealed moderate support for CAM modalities such as phytotherapy, manual therapies, and acupuncture. Participants demonstrated limited confidence in their knowledge of CAM. Although 72% acknowledged the importance of CAM training, only a minority demonstrated adequate knowledge of reimbursements and relevant legislation. Women and urban practitioners showed tendencies toward higher confidence and engagement with CAM compared to men and rural practitioners, as indicated by effect size results. **Conclusions:** This study highlights the need for focused, evidence-based CAM training programs to bridge knowledge gaps and enhance integration into primary care. Although constrained by factors such as a limited participant pool, the results of the small-scale study offer perspectives regarding CAM training and its significance in meeting the changing requirements of both health care providers and their clients.

## 1. Introduction

Complementary and alternative medicine (CAM) encompasses diverse medical and health practices that are not typically part of conventional medicine, but are used alongside (complementary) or instead of (alternative) standard treatments. CAM includes a wide range of modalities, such as phytotherapy, acupuncture, manual therapies (e.g., osteopathy, chiropractic), homeopathy, and neural therapy, among others. While some CAM approaches have evidence-based support, others remain controversial within the medical community [1,2,3]. Understanding how general practitioners (GPs) perceive and integrate these methods is essential for aligning medical education and clinical practice with evolving patient preferences and scientific standards. Its increasing acceptance among patients has underscored the necessity of evidence-based training and education to ensure that healthcare practitioners are equipped with the requisite knowledge and competencies to effectively integrate CAM into their clinical practice [4,5]. Incorporating CAM into medical curricula significantly enhances students’ openness and ability to integrate these modalities into future practice [6,7].

GPs play a pivotal role in primary care and often serve as the first point of contact for patients seeking CAM treatments [4,8]. Understanding their perspectives on CAM is crucial, especially in South Tyrol. The selection of this region is justified by its cultural and linguistic diversity, as well as the coexistence of conventional and CAM practices [9]. This context provides an ideal setting to explore the integration of CAM into medical training, ensuring that educational approaches align with professional standards and patient needs.

Despite the increasing prominence of CAM, there remains a lack of data regarding the attitudes and practices of GPs, particularly those who have recently completed their training [10]. Current studies indicate a considerable variation in GPs’ understanding, perspectives, and self-assurance regarding CAM. Whilst some GPs strongly endorse these practices, others remain doubtful, citing potential risks and a lack of robust scientific evidence to support their efficacy [5]. The inconsistency in CAM practices presents obstacles when developing standardized, evidence-based CAM education within medical programs. Furthermore, whilst the incorporation of CAM into healthcare is becoming more widely acknowledged, research suggests that numerous GPs feel ill-equipped to critically assess CAM methods or engage in discussions about them with patients [11,12,13]. It is essential to address these shortcomings to create curricula that not only support scientific proficiency and evidence-based approaches to CAM, but also provide GPs with the requisite skills to manage patient expectations and adapt to emerging integrative healthcare frameworks.

The integration of CAM into medical education requires a structured framework to ensure training is comprehensive, evidence-based, and aligned with clinical practice. Kern’s six-step approach to curriculum development provides a systematic method for embedding CAM training into GP education [14]. Applying this model, CAM education for GPs should focus on enhancing scientific literacy, critical appraisal of CAM research, and effective patient communication.

This study aimed to assess the attitudes, practices, and educational needs of GPs regarding CAM in South Tyrol. Specifically, it sought the following:Evaluate the level of support for various CAM modalities.Investigate the clinical applications and personal use of CAM among GPs.Examining the importance of scientific evidence and competence in CAM practices.Provide insights for curriculum development to better prepare future healthcare professionals for integrating CAM into their practice.

By focusing on early career general practitioners who are either currently in training or have completed their training within the past decade, this study investigated the evolving role of CAM in general practice education.

## 2. Methods

### 2.1. Study Setting and Participants

This study was conducted in South Tyrol, an autonomous province in northern Italy known for its cultural and linguistic diversity, with both German and Italian being recognized as official languages. The province’s unique healthcare system integrates traditional and modern medical practices, making it an ideal context for examining CAM in primary care.

The Institute of General Practice and Public Health is responsible for the specialization of GPs in South Tyrol. This study was designed as a complete population survey targeting all general practitioners who completed their specialization in South Tyrol within the last ten years (*n* = 131). As the objective was to assess the attitudes and training needs of this entire cohort, no a priori sample size calculation was performed. To minimize response bias, all GPs who specialized in South Tyrol within the preceding decade were invited to participate. Recruitment was conducted via email.

### 2.2. Survey Instrument

The questionnaire was adapted from a previous study in Germany [15], and a thesis performed in South Tyrol [16] was designed to assess GPs’ attitudes, knowledge, and practices regarding CAM. It comprises three sections:General Attitudes and Knowledge (Likert Scale Questions): This section included 20 items rated on a 6-point Likert scale (0 = “does not apply” to 5 = “fully applies”). Key topics include interest in CAM, perceived benefits and scientific evidence for CAM, integration of CAM into daily practice, perceived competence in addressing CAM-related patient questions and interactions, and experiences with and recommendations for CAM modalities. While the original German questionnaire was based on a content validation process involving input from GPs to align the items with relevant educational and clinical CAM topics, no formal psychometric testing (e.g., construct validity, reliability analysis) was reported [15]. For this study, the questionnaire was adapted to the local context using a structured translation workflow. The Italian version was first created using an AI-assisted translation tool (DeepL) and subsequently reviewed and refined by native-speaking Italians, including two practicing GPs, to ensure linguistic precision and cultural appropriateness. Given the exploratory and descriptive nature of this study and the complete target population approach, no formal psychometric validation was conducted on the Italian version. However, care was taken to preserve the original content intent and clarity across both languages used.Specific Practices and Experiences: Adapted from the thesis, this section explores CAM methods perceived as scientifically valid, personal and clinical applications of CAM, methods endorsed to patients, and preferences for future training and education in CAM [16].The demographic section covered age, sex, current professional status (in training or completed general practice specialization), workplace location (urban or rural and specific health district in South Tyrol), and years of practice since training completion. Employment type (full-time or part-time), additional specialist qualifications, and plans to continue working in South Tyrol were documented.

The questionnaire was distributed in German and Italian to accommodate South Tyrol’s multilingual context. The English translation is provided in Appendix A.

### 2.3. Data Collection

The survey was conducted anonymously to facilitate frank responses. The questionnaire was designed with neutral wording to avoid leading answers.

The online survey was available from 30 November 2024 to 31 January 2025. To enhance response rates, all 131 invited GPs received a follow-up email reminder on 2 January 2025. Additionally, verbal reminders were given during seminars attended by GPs in training, reinforcing the importance of participation and increasing engagement.

Data quality control measures included screening responses for completeness and internal consistency. Implausible or inconsistent entries were cross-checked against available background information for verification. Before statistical analysis, a final review was conducted to ensure the accuracy and reliability of the dataset.

### 2.4. Statistical Analysis

Frequencies and percentages were calculated to summarize categorical data, while medians (25th–75th percentiles) or medians (interquartile range) were reported where appropriate. The Shapiro–Wilk test was conducted to assess the distribution of the data, revealing deviations from normality for most variables. Consequently, non-parametric tests were used for comparisons.

For group comparisons, Mann–Whitney U tests were used for two-group comparisons (gender, practice location), while Kruskal–Wallis H tests were used for comparisons among three or more groups (mother tongue, country of medical studies). The effect sizes were calculated to evaluate the magnitude of the observed differences.

Rank-Biserial Correlation (r) was calculated using Mann–Whitney U tests and interpreted as small (r < 0.3), moderate (r = 0.3–0.5), or large (r > 0.5).Eta-squared (η^2^) was calculated for Kruskal–Wallis H tests and interpreted as small (η^2^ < 0.10), medium (η^2^ = 0.10–0.30), or large (η^2^ > 0.30), based on an adaptation of Cohen’s guidelines for non-parametric data [17].

Cross-comparisons with previous studies conducted in South Tyrol were qualitatively discussed to contextualise the findings and identify trends in attitudes and practices regarding CAM. When interpreting results, effect sizes were considered alongside *p*-values to identify potential trends. In line with recent recommendations, moderate-to-large effect sizes were interpreted as indicating potentially meaningful differences, even in the absence of statistical significance, particularly considering the limited sample size [18,19].

## 3. Results

### 3.1. Characteristics of the Study Participants

Of the 131 invited GPs, 38 responded to the survey, with 37 evaluable responses. The participants had diverse demographic and professional backgrounds, as summarized in Table 1. Most respondents were female (65%), with a significant proportion aged >40 years (43%). The native languages were predominantly German (46%) and Italian (51%), reflecting the bilingual context of South Tyrol, with one participant identifying Ladin as their mother tongue. Over half of the respondents reported being married or in partnerships.

Regarding education, most participants completed their medical studies in Italy (51%) or Austria (35%), with the remainder studying in Germany. Professionally, 66% of the participants were trained in GPs, with the remaining still in training, including four trained specialists undergoing further general practice training. Most respondents (83%) were actively working in general practice, 27% were based in rural areas, and 51% practiced in urban locations. The remaining 22% had no dedicated practice, often reflecting training.

### 3.2. Perspectives, Practices, Knowledge, and Training in CAM

The results indicated a clear demand for evidence-based education and structured CAM training among GPs. Despite favorable attitudes towards CAM integration, practical use of and confidence in CAM knowledge were limited, highlighting the need for tailored educational programs to bridge the knowledge gap and improve CAM integration into general practice.

Table 2 shows that most participants had a moderate interest in CAM (mean score 2.5), with only 19% reporting a strong interest and 29.7% expressing no interest. Respondents were more positive about CAM integration into primary care, with 30% fully agreeing that it benefited patients. A significant emphasis was placed on evidence, with 54% indicating it as a prerequisite for CAM application, with the highest mean score in this category (mean score 3.9).

CAM use in daily practice was limited, with low mean scores (1.8) for applying CAM and asking patients about their CAM use (mean score 1.8). Only 14% of participants reported regular CAM use in practice, and 6% attended additional CAM training. Interestingly, 8% planned future CAM training, and 16% used CAM personally, despite low professional application levels.

Respondents rated their CAM knowledge and confidence as limited. Only 5% felt their knowledge was sufficient (mean 1.5, SD 1.32), and confidence in evaluating interactions, evidence, or media information was low (mean scores 1.9–2.2). Confidence in addressing legal and reimbursement questions was particularly low (mean scores 1.4 of 1.6). Meanwhile, 14% expressed a strong interest in CAM courses within GP training, and 17% attended CAM-related courses.

Figure 1 illustrates the key findings on GPs’ attitudes, practices, and confidence regarding CAM. The stacked bar chart highlights the proportions of participants who expressed strong agreement (Likert 5–6) or strong disagreement (Likert 1–2) across various aspects, including CAM interest, application in practice, confidence in knowledge, and training preferences. Notably, while many participants acknowledged the importance of CAM and its evidence base, confidence in practical application and legal aspects remained low.

### 3.3. Comparisons of CAM-Related Attitudes and Practices

Statistical analyses revealed one significant difference in CAM-related practices between men and women. Female physicians were significantly more likely than their male counterparts to actively ask their patients whether they use CAM (*p* = 0.045, r = 0.40; Appendix A).

For all other comparisons, no significant differences were observed across sex, native language, country of medical studies, or practice location (all *p* > 0.05). However, some comparisons showed moderate effect sizes (r = 0.3–0.5) suggesting meaningful trends. For instance, women tended to report greater interest in CAM courses during GP training and a greater inclination to integrate CAM into primary care (r ≈ 0.31). Additionally, urban GPs reported slightly higher confidence in evaluating CAM reimbursement questions compared to their rural counterparts, though this effect was small (r ≈ 0.24).

Comparisons based on native language and country of medical studies demonstrated small effect sizes (η^2^ < 0.10), indicating minimal practical differences in CAM-related attitudes and practices among these groups.

Overall, these findings suggest that while CAM-related attitudes and practices appear largely uniform across primary care physicians, notable tendencies exist regarding gender differences in patient communication and slight variations by practice location. Women were more likely to engage with patients about CAM, and urban GPs exhibited slightly greater confidence in navigating CAM reimbursement issues. However, these effects were modest, and further research is needed to clarify their practical significance.

### 3.4. Attitudes and Practices Among Participants Regarding CAM Methods

#### 3.4.1. Overview of Attitudes and Practices

Table 3 provides an overview of the participants’ attitudes and practices regarding CAM methods. Phytotherapy is the most supported method, is widely considered scientifically valid, is frequently recommended to patients, and is commonly used personally or within families. Manual therapy, osteopathy, and chiropractic therapy ranked second across most categories, showing strong support and application in both the clinical and personal contexts.

Acupuncture had high levels of perceived scientific validity and support, but was less commonly recommended to patients or used personally. Folk medicine/home remedies and neural therapy had moderate levels of support and application, although these methods were perceived less often as scientifically valid.

Homeopathy had the lowest levels of support and perceived scientific validity, though a small proportion of participants reported positive experiences and personal or family use. Interestingly, a subset of participants did not support any CAM method, with a larger proportion indicating that they did not personally use any of the listed methods.

#### 3.4.2. Indications for CAM Prescriptions

Participants were asked to indicate the types of complaints for which they had prescribed CAM. On average, participants provided 4.2 responses to this item, with a median of four responses and a range of one to six. Musculoskeletal complaints were the most frequently mentioned indication and were cited by 56% of the participants. This was followed by psychosomatic complaints (52%) and asthenia, weakness, and fatigue (40%). Unclear complaints without pathological findings were reported by 36% of patients, and headaches were cited by 32%. A small minority of participants (8%) indicated that they did not prescribe CAM for complaints.

#### 3.4.3. Situations for Recommending CAM

Participants were also asked about situations in which they recommended CAM. On average, the participants provided 1.8 responses to this item, with a median of 2 responses and a range of 1 to 3 responses per participant. The most frequent response was recommending CAM as a complement to conventional medicine, mentioned by 71% of the participants. Recommending CAM at the patient’s request was the second most common response, reported by 54%. A small proportion (4%) indicated that they recommended CAM instead of conventional medicine, whereas 21% stated that they never recommended CAM.

#### 3.4.4. CAM Educational Background

Another question was whether the participants had learned about the CAM method. On average, the participants provided 1.3 responses, with most indicating a single method or response. The most common response was “No”, mentioned by 34% of participants. Additionally, 17% indicated that they had not learned the CAM method but expressed interest in doing so. Among those who had learned the CAM method, acupuncture was the most frequently mentioned (10%), followed by neural therapy (7%). Other methods, such as homoeopathy, manual therapy, phytotherapy, and taping, were mentioned by smaller proportions of participants. A small subset (6.9%) reported that they were currently learning the CAM method.

#### 3.4.5. CAM in Clinical Practice and Future Perspectives

Most participants (34.5%) were not interested in offering CAM in their practice, while 28% cited a lack of time or other reasons for not offering CAM. However, 24% expressed an interest in offering CAM in the future. Among current CAM providers, acupuncture and neural therapy are specifically mentioned by smaller subsets. A minority (3%) indicated current offerings and future aspirations.

When asked whether it was sufficient for GPs to refer patients to CAM specialists, 48% believed that GPs should not prescribe CAM themselves. Conversely, 34% felt that GPs should offer some CAM methods personally, and 34% agreed that referrals to CAM specialists were sufficient. A minority provided mixed responses, indicating agreement with the need for personal CAM practices by GPs.

The participants were asked to assess their CAM knowledge. The majority (48%) rated their knowledge as “Limited”. A significant proportion (31%) described their knowledge as “Good”, while smaller percentages assessed it as “Insufficient” (14%) or “Excellent” (3%). These findings suggest that while a subset of participants felt confident in their CAM knowledge, most perceived it as either limited or in need of improvement.

The participants were also asked about the perceived usefulness of CAM training for GPs. The majority (72%) considered such training to be valuable, with 38% rating it as “very useful” and 34% as “somewhat useful”. In contrast, 28% of participants viewed CAM training as “not useful”. These findings suggest that, while most participants recognize the importance of CAM training for general practitioners, a smaller subset remains skeptical about its utility in primary care settings.

Finally, participants were asked whether CAM should receive financial support from public funds. A minority (17%) responded positively with “Yes”. Additionally, 24% of participants agreed partially, selecting “Mostly yes”. The most frequent response, however, was “Rarely”, mentioned by 34% of participants. Meanwhile, 24% expressed opposition to public funding for CAM, responding with “Never”. These findings indicate a wide spectrum of opinions on public funding for CAM, ranging from full support to outright opposition, with many participants expressing conditional or limited approval.

## 4. Discussion

This study examines the views, approaches, and training requirements of GPs regarding CAM in South Tyrol. This investigation describes how GPs in the region perceive and utilize CAM while also highlighting their educational necessities in this area. While there is substantial support for certain CAM modalities, particularly phytotherapy, manual therapy, osteopathy, and chiropractic therapy, their practical use in daily practice remains limited. The findings suggest a notable gap between favorable attitudes towards CAM and participants’ confidence and knowledge levels, with the majority rating their knowledge as either limited or insufficient. Despite these challenges, participants demonstrated a strong emphasis on evidence-based practice and showed a significant interest in structured CAM training, with more than two-thirds perceiving such training as valuable. However, opinions on CAM’s integration into healthcare systems were divided, with contrasting views on public funding and the appropriate role of GPs in providing or referring to CAM. These findings underscore the need for tailored educational programs and clearer policy frameworks to bridge knowledge gaps and enhance the integration of CAM into general practice.

A key limitation of this study is its low participation rate, with only 37 of 131 invited young-career GPs in South Tyrol responding despite a reminder. This response rate underscores the challenge of obtaining robust insights from a small, specialized population. GP training in South Tyrol is unique and limited in scale, making it difficult to expand the study population. The low response rate may reflect the controversial nature of CAM, which remains polarizing in medicine. Some GPs may have refrained from participating due to differing attitudes, skepticism, or lack of engagement with CAM, potentially introducing selection bias. Despite these limitations, sharing these results in the medical education literature is justified.

This study addresses CAM integration into general practice education in a multilingual multicultural region. South Tyrol’s bilingual, bicultural context offers unique insights into CAM’s evolving role in primary care education, providing lessons for similar regions. The findings enhance understanding of GPs’ educational needs regarding CAM, revealing deficiencies in confidence, knowledge, and evidence-based application. This is timely due to growing global interest in integrating CAM into primary care. This study emphasizes the necessity of tailored educational programs to address these gaps, supporting efforts to incorporate evidence-based CAM into medical curricula.

Despite the low response rate limiting generalizability, respondents likely represented an engaged group interested in CAM, providing valuable insights. Publishing these results will initiate future research and discussion in medical education, promoting more comprehensive studies. This study addresses a critical, underexplored topic, contributing to the literature on CAM and primary care education, and lays the groundwork for further research.

The findings in South Tyrol were compared with those reported by Ref. [15], who surveyed GPs in training across Germany to assess their attitudes, practices, and educational needs regarding CAM. The German study offers a useful comparison due to its similar focus on early career GPs and use of a published questionnaire also employed in the current study. Both studies explored CAM integration into primary care and identified educational and practical gaps, revealing regional similarities and differences.

GPs in both studies demonstrated positive attitudes toward CAM, with greater support for phytotherapy, acupuncture, and manual therapies. South Tyrolean participants emphasized evidence as essential for CAM use (mean score 4.1) more than German counterparts, indicating regional priorities in scientific validation. Despite favorable attitudes, CAM use was limited in both studies. In South Tyrol, only 7% of participants reported regular CAM use, like low application rates in Germany. Barriers included time constraints and limited training; however, interest in CAM training was high. In South Tyrol, 73% of participants valued CAM training, aligning with German GPs who wanted CAM in the general practice curriculum [15].

South Tyrolean GPs reported less confidence in CAM knowledge and application than Germans, indicating a greater need for targeted educational interventions. The cultural and systemic context, including South Tyrol’s bilingual and multicultural healthcare system and Germany’s structured national CAM guidelines, may explain these differences. Both studies highlight the necessity of addressing educational and systemic barriers to CAM integration. These findings emphasize the need for evidence-based training to help GPs incorporate CAM into primary care, contributing to the ongoing discourse on CAM in medical education.

The findings of this study in South Tyrol align with Marcher’s (2017) thesis surveying general GPs in Bressanone/Brixen and Brunico/Bruneck districts [16]. In Marcher’s study, 90% supported at least one CAM method, favoring acupuncture, manual therapy, and phytotherapy. The current study also favored these methods, but with lower support (79% for phytotherapy). This difference may stem from Marcher’s more experienced GPs compared to younger practitioners in the current study. In Marcher’s study, 58% offered CAM, versus 7% in the current study, likely due to experience disparity. Both studies showed significant interest in CAM training, with 72% valuing it in the current study. Confidence in CAM knowledge varied: 74% of Marcher’s respondents rated their knowledge as “good” or better, compared to 31% in the current study, where most rated it as “limited” or “insufficient”. Both studies highlighted the need for structured CAM training to improve integration into primary care. Marcher’s findings offer a perspective from a more experienced cohort, while the current study provides insights into younger GPs’ attitudes and practices, emphasizing the importance of tailored educational initiatives.

In Switzerland, a significant portion of primary care physicians either offer CAM services themselves or refer patients to CAM treatments, with homeopathy, phytotherapy, and traditional Chinese medicine/acupuncture being the most common [18,19]. Scandinavian patients with conditions like cancer or multiple sclerosis often use CAM as a complement to conventional treatments. They value both scientific and experience-based knowledge, which aligns with the patient-centered goals of European public health programs [20]. Initiatives to train general practitioners in incorporating CAM into their clinical practice began several years ago. Educational programs have been established to assist doctors in creating personalized treatment strategies that include CAM, thereby enhancing their capacity to address patients’ requirements [21,22].

This study on South Tyrolean GPs revealed differences in CAM preferences according to sex and practice location (Appendix A). Women were more likely than men to view CAM positively and feel more confident in evaluating CAM-related evidence and patient enquiries. The lack of statistical significance in the observed differences in CAM attitudes and practices between groups defined by sex and practice location in this study could be attributed to the small sample size, which limits statistical power. However, the effect size analyses reveal moderate to large effects (r = 0.4–0.7), suggesting meaningful trends. As highlighted by Ref. [23], the absence of statistical significance should not be misinterpreted as evidence of no effect. Instead, the effect sizes provide valuable insights into potential differences that warrant further exploration in larger studies. Ref. [24] also underscores the importance of interpreting effect sizes alongside *p*-values to avoid over-reliance on binary thresholds of statistical significance, which can obscure meaningful findings. In this context, the observed effect sizes indicate tendencies, such as women and urban practitioners exhibiting higher confidence and engagement with CAM. These findings are consistent with those of a previous study that indicated significant sex differences in CAM use in the South Tyrolean population [9]. The population-based study found that women were significantly more likely than men to consult CAM providers such as homoeopaths, acupuncturists, and osteopaths, mirroring the current findings among GPs. In addition, urban residents and individuals with higher education levels showed a greater inclination towards CAM, paralleling the higher confidence levels among urban GPs in evaluating CAM-related aspects. Urban practitioners exhibited slightly higher confidence in assessing CAM reimbursement issues than did rural practitioners.

The observed alignment between GPs’ preferences of GPs and those of the general population suggests that healthcare professionals’ preferences may reflect broader societal trends in South Tyrol. This underscores the need for tailored CAM integration approaches that address a region’s cultural, linguistic, and educational dynamics. Such strategies could better align GPs’ practices with patient expectations, thereby enhancing the effectiveness and acceptance of CAM in primary care settings.

### 4.1. Strengths and Limitations

This study’s small sample size, previously noted as a significant limitation, affected the generalizability of the findings. The low participation rate among young career GPs in South Tyrol may have introduced a selection bias, reflecting respondents with stronger opinions or interests in CAM. A limitation is the inability to characterize non-respondents due to survey anonymity. However, as all invited participants were young GPs specialized in South Tyrol within the last 10 years, their demographic and professional backgrounds were likely similar. Non-response bias remains possible, with those more engaged in CAM potentially being overrepresented. As a result, we cannot rule out selection bias, particularly if GPs with a greater interest in CAM were more likely to respond. Additionally, the cross-sectional design restricted causal inferences between attitudes, practices, and educational needs. Self-reporting bias was possible, with participants potentially overestimating or underestimating their CAM knowledge or use. The predefined questionnaire may have limited the exploration of less common or emerging CAM modalities. The unique sociocultural and healthcare context of South Tyrol, with its bilingual and multicultural environment, may further limit the transferability of these findings to other regions.

Nonetheless, this study had several strengths. This study provides an in-depth exploration of CAM attitudes, practices, and educational needs among South Tyrol GPs, a region with a distinctive healthcare model and cultural context. Focusing on young career GPs offers insight into the evolving role of CAM in primary care education. The use of a standardized questionnaire, previously applied in other studies, enhances the comparability of findings across regions. Supplementary analyses of sex and practice location differences provide a nuanced understanding of CAM integration challenges and opportunities. These strengths highlight the relevance of this study to ongoing discussions on CAM in medical education, offering a foundation for future research and policy development.

### 4.2. Practical Implications

Given the identified gaps in CAM knowledge, confidence, and practical application among young-career GPs, these findings strongly support the inclusion of structured CAM modules in medical education. The high level of interest in CAM training supports a clear demand for targeted educational initiatives. Integrating evidence-based CAM training into GP residency programs could enhance physicians’ ability to critically evaluate CAM treatments, address patient inquiries with confidence, and understand legal and reimbursement frameworks. Furthermore, given the higher engagement of female GPs in discussing CAM with patients, training should emphasize communication strategies for integrating CAM discussions into primary care consultations. Educational approaches should also consider regional differences in healthcare practices, ensuring that training aligns with both scientific rigor and local patient expectations. Addressing these educational gaps through structured curricula and continuing medical education (CME) programs could improve the responsible integration of CAM into primary care, ultimately benefiting both practitioners and patients.

Implementing standardized curricula, interdisciplinary workshops, and CME initiatives can enhance GP competence. Integrating CAM into primary care requires structured training modules aligned with evidence-based medicine. Clear referral guidelines, reimbursement structures, and policy frameworks would support safe CAM use. Collaboration between medical institutions, policymakers, and CAM practitioners is essential to ensure effective implementation.

## 5. Conclusions

This study highlights the attitudes, practices, and educational needs of young career GPs in South Tyrol regarding CAM. While modalities such as phytotherapy, manual therapies, and acupuncture have received notable support, their practical use and application in daily practice remain limited. A significant gap was identified between favorable attitudes towards CAM and confidence in its application, underscoring the importance of targeted evidence-based CAM training programs.

The preferences for CAM among different genders and geographical locations in South Tyrol mirrored the trends observed in the broader population. Female practitioners and those in urban settings exhibited a greater inclination towards CAM usage, which aligns with the wider societal patterns. This underscores the importance of developing approaches to CAM integration that are sensitive to cultural and regional variation.

To translate these findings into practice, medical education programs should incorporate structured CAM modules into both undergraduate and postgraduate training. These modules should:Prioritize evidence-based content and critical appraisal of CAM literature;Include practical case-based learning on evaluating CAM efficacy, safety, and patient communication;Address legal, regulatory, and reimbursement frameworks relevant to CAM;Be integrated longitudinally, not limited to electives, to reflect CAM’s growing role in patient care.

Despite this study’s limitations, including a small sample size and potential selection bias, this study provides a deeper understanding of the challenges and opportunities in integrating CAM into primary care. The findings emphasize the need to address systemic and educational barriers and align GP training with evidence-based practices to meet the growing interest in CAM. Beyond addressing educational gaps, integrating CAM into primary care requires clear, evidence-based clinical guidelines to support GPs in safely incorporating CAM modalities into patient care. The absence of standardized recommendations may lead to uncertainty in practice and inconsistent patient counseling. Developing structured guidelines that define indications, contraindications, risks, and an evidence base of common CAM methods would ensure their responsible application in general practice. These guidelines should be developed collaboratively by medical associations, regulatory bodies, and CAM experts, aligning with scientific evidence and regional healthcare policies. Establishing clear best-practice frameworks can enhance patient safety, physician confidence, and CAM integration into evidence-based medicine.

This study serves as a foundation for further research and policy initiatives aimed at optimising CAM integration into healthcare education and practice.

## Figures and Tables

**Figure 1 healthcare-13-00797-f001:**
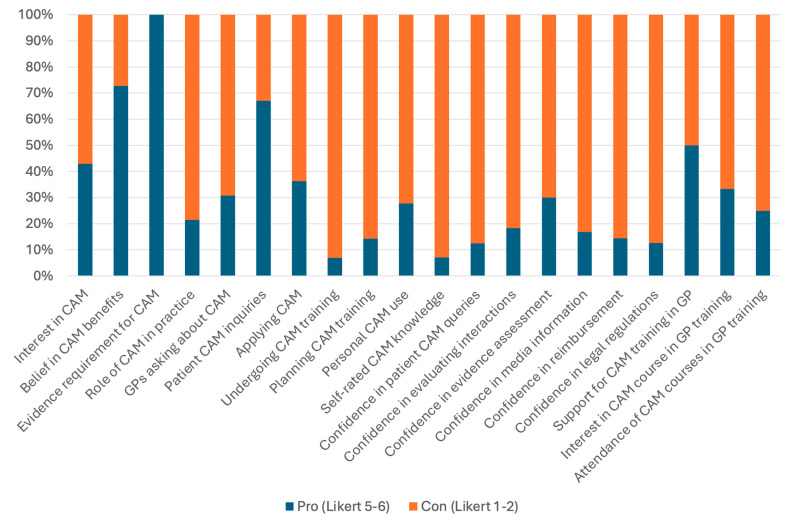
GPs’ attitudes, practices, and confidence regarding complementary and alternative medicine (CAM).

**Table 1 healthcare-13-00797-t001:** Description of the study population.

Variable		Participants, *n* = 37 (%)
Age (years)	<30	4 (10.8)
	31–35	8 (21.6)
	36–40	9 (24.3)
	>40	16 (43.2)
Sex	Male	13 (35.1)
	Female	24 (64.9)
Native Language	German	17 (45.9)
	Italian	19 (51.4)
	Ladin	1 (2.7)
Medical Studies Location	Italy	19 (51.4)
	Austria	13 (35.1)
	Germany	5 (13.5)
Marital Status	Married/partnership	22 (59.5)
	Single	12 (32.4)
	Divorced	3 (8.1)
Specialization Status as GP	In training ^1^	14 (37.8)
	Trained	23 (62.2)
Work in General Practice	Yes ^2^	30 (81.1)
	No ^3^	7 (18.9)
Activity as GP (years)	In training	14 (37.8)
	<4	6 (16.2)
	4–7	9 (24.3)
	>7	8 (21.6)
Practice Location	Rural	10 (27.0)
	Urban ^4^	19 (51.4)
	Not applicable	8 (21.6)

^1^ Including five trained specialists in general practice training. ^2^ Including four participants in training as GP. ^3^ Including one trained GP. ^4^ Bolzano/Bozen, Merano/Meran, Bressanone/Brixen, or Brunico/Bruneck.

**Table 2 healthcare-13-00797-t002:** Attitudes, practices, knowledge, and training in complementary and alternative medicine among general practitioners.

Category	Question	Participants (*n*)	Mean (SD)	Median (25th; 75th Percentiles)	Does Not Apply/Never (%)	Fully Applies/Very Often (%)
Attitudes Toward CAM	I am interested in the topic of CAM	37	2.54 (1.79)	2 (1; 4)	29.7	18.9
	The integration of CAM into primary care benefits patients	37	2.68 (1.87)	3 (1; 5)	13.5	29.7
	Adequate evidence is a prerequisite for applying CAM	37	3.89 (1.35)	4 (3; 5)	0.0	54.1
Use of CAM	CAM plays an important role in my daily work	37	1.84 (1.52)	2 (0; 3)	37.8	10.8
	I actively ask my patients whether they use CAM	37	1.81 (1.61)	2 (0; 3)	32.4	13.5
	My patients ask me for advice regarding CAM	36	1.67 (1.45)	2 (1.75; 4)	5.4	8.1
	I apply CAM in my medical practice	37	1.73 (1.58)	2 (0; 3)	24.3	13.5
	I am currently undergoing additional training in a CAM method	36	0.89 (1.78)	0 (0; 0)	86.1	5.6
	I plan to pursue additional training in CAM	37	1.46 (1.98)	0 (0; 3)	58.3	8.3
	I use CAM personally	37	1.57 (1.82)	1 (0; 2)	43.2	16.2
Knowledge in CAM	I consider my knowledge of CAM sufficient	37	1.49 (1.32)	1 (1; 2)	40.5	5.4
	I feel confident when patients ask me about CAM	37	2.00 (1.43)	2 (1; 3)	24.3	5.4
	I feel confident evaluating potential interactions, side effects, and contraindications in CAM	37	1.92 (1.44)	2 (1; 3)	27.0	8.1
	I feel confident evaluating the evidence base for CAM	37	2.19 (1.61)	2 (1; 3)	21.6	10.8
	I feel confident evaluating CAM-related information in lay media	37	1.92 (1.60)	1 (1; 3)	32.4	8.1
	I feel confident regarding CAM reimbursement questions	36	1.58 (1.57)	1 (0; 3)	36.1	8.3
	I feel confident regarding CAM legal regulations	37	1.35 (1.46)	1 (0; 3)	48.6	8.1
Training in CAM	GP training should include competencies in CAM	37	2.68 (1.87)	3 (1; 4)	16.2	21.6
	I am interested in a CAM course in the GP training program	37	2.49 (1.84)	3 (1; 4)	27.0	13.5
	I have already attended CAM-related courses as part of GP training of GPs	37	1.73 (2.08)	0 (0; 4)	50.0	16.7

Means and medians are based on a scale from 0 to 5 (0 = does not apply/never; 5 = fully applies/very often). Abbreviations: GP, general practitioner; CAM, complementary and alternative medicine; SD, standard deviation.

**Table 3 healthcare-13-00797-t003:** Summary of CAM support, perceived scientific validity, positive experiences, methods recommended to patients, and methods used personally or by family (*n* = 29).

CAM Method ^1^	Support *n* (%)	Considered Scientifically Valid *n* (%)	Positive Experiences *n* (%)	Recommended to Patients *n* (%)	Used Personally or for Family *n* (%)
Phytotherapy	23(79)	22 (76)	17 (59)	18 (62)	16 (55)
Manual Therapy, Osteopathy, Chiropractic	16 (55)	13 (45)	11 (38)	14 (48)	11 (38)
Acupuncture	20 (69)	20 (69)	17 (59)	12 (41)	6 (21)
Folk Medicine, Home Remedies	12 (41)	5 (17)	9 (31)	9 (31)	9 (31)
Neural Therapy ^2^	11 (38)	6 (21)	4 (14)	6 (21)	4 (14)
Homeopathy	6 (21)	3 (10)	6 (21)	1 (3)	4 (14)
None	3 (10)	2 (7)	4 (14)	6 (21)	13 (45)

^1^ The listed methods were offered as predefined response options, with an additional free-text field available for participants to suggest other methods. No additional methods were used in the present study. ^2^ Neural therapy involves the use of local anesthetics, typically procaine, to diagnose and treat autonomic nervous system dysfunction and relieve chronic pain. Abbreviations: CAM, complementary and alternative medicine.

## Data Availability

Data are available from the corresponding author upon reasonable request.

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
