# Peer review of "Integrating Complementary and Alternative Medicine into General Practice Training: A Regional Survey in South Tyrol"

_healthcare, 2025, doi:10.3390/healthcare13070797_

Round 1
Reviewer 1 Report (Previous Reviewer 1)
Comments and Suggestions for Authors
Evaluating the revisions made to the manuscript titled "Integrating Complementary and Alternative Medicine into General Practice Training: A Regional Survey in South Tyrol", I can confirm that the authors have appropriately considered the suggestions and made significant improvements to the clarity and coherence of the manuscript.
Title: The title remains appropriate for the study, accurately reflecting the research focus and the regional context addressed.
Abstract: The suggestions for the abstract were well incorporated, providing a more precise and clear description of the results. The revision of the phrase "practice and confidence in CAM knowledge were inadequate" to "Participants demonstrated limited confidence in their knowledge of CAM" more clearly conveys the limitations observed in the participants, while the revision about the recognition of the importance of CAM training offers better clarity on the gap identified regarding knowledge of reimbursements and legal aspects.
Introduction: The changes suggested for the introduction were relevant, particularly the revision of the phrase regarding the role of general practitioners (GPs), making it more accurate. Additionally, the justification for selecting South Tyrol was sufficiently elaborated, reinforcing the significance of the region's cultural and linguistic diversity and the coexistence of conventional and complementary medical practices.
Methods: The suggestion to specify whether the questionnaire used had prior validation was effectively incorporated, strengthening the methodological rigor of the study. The minor adjustment to the phrasing regarding the survey's dates improved the temporal clarity of the presentation.
Results: The revisions to the results, such as clarifying the regular use of CAM and participation in additional CAM training, made the data description more precise and understandable. The addition of a concluding paragraph that highlights the main differences between the analyzed groups (e.g., gender, practice location) is a valuable improvement for enhancing the comparative analysis and providing a better understanding of the data.
Discussion: The suggestions in the discussion were effectively implemented, with the phrase regarding GPs' attitudes toward CAM now being clearer. Emphasizing the practical implications of the results, particularly regarding the inclusion of specific CAM modules in medical training, strengthens the study's impact. The suggestion to include recommendations for integrating CAM into medical curricula was appropriately addressed, reflecting the study's practical relevance.
Conclusion: The conclusion was appropriately strengthened, emphasizing the need for the development of clear guidelines for integrating CAM into clinical practice, which is a pertinent and relevant recommendation for the field.
References: The references are up-to-date and appropriate for the context of the article.
Overall, the authors have appropriately addressed the suggested revisions, enhancing the clarity, coherence, and depth of the manuscript. The study is now more robust and substantiated, with a stronger argumentation and clearer presentation of the results. Therefore, I consider the manuscript ready for acceptance for publication.
Recommendation: Accept for publication.
Author Response
We thank Reviewer 1 for the constructive evaluation of our revised manuscript and for the positive recommendation for publication.
Reviewer 2 Report (Previous Reviewer 2)
Comments and Suggestions for Authors
In the introduction, line 13 of page 1 mentions 37 analyzed responses, but in the results, it refers to 31 responses on page 4, line 193. The tables again indicate the sample size as 37.
The sample size is not justified in the methodology. What was the minimum number required for the results to be consistent? Do the 31 or 37 analyzed responses allow for statistically significant conclusions?
Would it be possible to increase the sample size?
Author Response
Comment:
In the introduction, line 13 of page 1 mentions 37 analyzed responses, but in the results, it refers to 31 responses on page 4, line 193. The tables again indicate the sample size as 37.
The sample size is not justified in the methodology. What was the minimum number required for the results to be consistent? Do the 31 or 37 analyzed responses allow for statistically significant conclusions?
Would it be possible to increase the sample size?
Response:
The discrepancy in reported sample size (i.e., 31 vs. 37) was the result of a typographical oversight, which has now been corrected throughout the manuscript. As clarified in the revised version, 131 young-career GPs in South Tyrol were invited to participate in the survey. After one reminder, 32 responses were received, of which 31 were evaluable. Following a second recruitment effort, 6 additional responses were obtained, resulting in a total of 37 evaluable responses that form the basis of the final analysis.
This study was conceived as a complete population survey, targeting all general practitioners who completed their specialization in South Tyrol within the last ten years. As such, no a priori sample size calculation was performed. This has now been explicitly stated in the Methods section as follows:
“This study was designed as a complete population survey targeting all general practitioners who completed their specialization in South Tyrol within the last ten years (n = 131). As the objective was to assess the attitudes and training needs of this entire cohort, no a priori sample size calculation was performed. To minimize response bias, all GPs who specialized in South Tyrol within the preceding decade were invited to participate.”
Given the small and defined population of interest, and despite reminders and direct contact during teaching events, no further increase in sample size was possible. While we acknowledge that the response rate limits generalizability, the study nonetheless offers valuable exploratory insights into a specific and under-researched professional group in a bilingual, multicultural healthcare setting. This limitation has been transparently discussed in the Strengths and Limitations section.
Reviewer 3 Report (Previous Reviewer 3)
Comments and Suggestions for Authors
-
Discrepancy in sample size: The text states "32 responded to the survey with 31 evaluable responses" (p.4), while Table 1 and subsequent analyses indicate n=37. This significant inconsistency must be addressed throughout the manuscript.
Section 3.3 of the statistical analysis interpretation necessitates revision to more explicitly clarify the correlation between statistical significance and effect sizes. Presently, inconsistencies exist between the text and Supplementary Table S1. The approach used for interpreting "meaningful trends" without statistical significance requires stronger justification.
The Methods section should provide more on the adaptation and translation process of the questionnaire, specifically explaining the linguistic and cultural validation measures employed in addition to the AI-based translation.
The limitations section recognizes the low response rate (37/131); however, a comparative analysis of respondents versus the general GP population in South Tyrol would enhance the discussion on potential selection bias.
The Conclusions section should include more precise, actionable recommendations for the integration of CAM into medical curricula, in alignment with the study's findings.
Several grammatical and formatting inconsistencies require attention, particularly in tracked changes visible throughout the manuscript.
Author Response
Please see the attachment.

This manuscript is a resubmission of an earlier submission. The following is a list of the peer review reports and author responses from that submission.
Round 1
Reviewer 1 Report
Comments and Suggestions for Authors
Title: Is adequate to the study developed.
Abstract:
Original phrase: "practice and confidence in CAM knowledge were inadequate". Suggestion: "Participants demonstrated limited confidence in their knowledge of CAM".
Original phrase: "72% of participants recognized the value of CAM training, but knowledge of reimbursements and legal aspects was insufficient".
Suggestion: "Although 72% of participants acknowledged the importance of CAM training, only a minority demonstrated adequate knowledge of reimbursements and relevant legislation".
Introduction
Original phrase: "General practitioners (GPs) play a pivotal role in primary care and often serve as the first point of contact for patients seeking CAM treatments".
Suggestion: "General practitioners (GPs) play a crucial role in primary care, often serving as the first point of contact for patients interested in CAM".
The study should better justify the choice of South Tyrol.
Suggestion: "The selection of this region is justified by its cultural and linguistic diversity, as well as the coexistence of conventional and complementary medical practices".
Methods
Specify whether the questionnaire used had prior validation.
Original phrase: "The online survey was open from 30 November 2024 to 10 January 2025".
Suggestion: "The online survey was available from November 30, 2024, to January 10, 2025".
Results
Original phrase: "Only 6.9% reported regular CAM use in practice, and 10.3% underwent additional CAM training".
Suggestion: "Only 6.9% of participants reported regular CAM use in clinical practice, while 10.3% attended additional CAM training".
Add a final paragraph reinforcing the main differences among analyzed groups (gender, practice location, etc.).
Discussion
Original phrase: "GPs in both studies showed positive attitudes towards CAM, with phytotherapy, acupuncture, and manual therapies most supported".
Suggestion: "GPs in both studies demonstrated positive attitudes toward CAM, with greater support for phytotherapy, acupuncture, and manual therapies".
Strengthen the practical implications of the results, suggesting the inclusion of specific CAM modules in medical training.
Conclusion
Emphasize the need to develop clear guidelines for integrating CAM into clinical practice.
References: Adequate and actual.
Reviewer 2 Report
Comments and Suggestions for Authors
I think that what is meant by alternative and complementary medicine is not well defined and it would be worthwhile to define this fundamental part of the study.
I also believe that the sample size is insufficient to draw valid conclusions and that an attempt should be made to recruit a larger sample size.
Reviewer 3 Report
Comments and Suggestions for Authors
Introduction
- Include more recent references from 2023-2024 regarding the integration of complementary and alternative medicine (CAM) in medical education.
- Enhance the justification for focusing on South Tyrol.
- Enhance the clarity of the research gaps in the literature on CAM education.
- Incorporate a clear theoretical framework for the integration of complementary and alternative medicine (CAM) within medical education.
Methods
- Provide more details on the process of questionnaire translation and its validation.
- Elucidate the methods employed to mitigate response bias.
- Elucidate the reminder procedure for non-respondents.
- Describe data quality control measures.
Results
- Consider adding a figure or infographic to make the manuscript more attractive to readers.
- Standardize the reporting of percentages.
- Clarify the characteristics of non-respondents.
Discussion
- Enhance the comparison with international literature.
- Elaborate on the implications for medical education policy.
- Provide suggestions for curriculum development.
- Discuss possible implementation strategies.
Comments on the Quality of English Language
- It is advisable to seek professional editing services.
- Enhanced precision in topic sentences
- Improved transition between sections
